# Thermostable virus portal proteins as reprogrammable adapters for solid-state nanopore sensors

Benjamin Cressiot [1,2,5], Sandra J. Greive [3], Mehrnaz Mojtabavi[4], Alfred A. Antson[3] & Meni Wanunu [1,2]

Nanopore-based sensors are advancing the sensitivity and selectivity of single-molecule detection in molecular medicine and biotechnology. Current electrical sensing devices are based on either membrane protein pores supported in planar lipid bilayers or solid-state (SS) pores fabricated in thin metallic membranes. While both types of nanosensors have been used in a variety of applications, each has inherent disadvantages that limit its use. Hybrid nanopores, consisting of a protein pore supported within a SS membrane, combine the robust nature of SS membranes with the precise and simple engineering of protein nanopores. We demonstrate here a novel lipid-free hybrid nanopore comprising a natural DNA pore from a thermostable virus, electrokinetically inserted into a larger nanopore supported in a silicon nitride membrane. The hybrid pore is stable and easy to fabricate, and, most importantly, exhibits low peripheral leakage allowing sensing and discrimination among different types of biomolecules.

---

[1] Department of Physics, Northeastern University, Boston, MA 02115, USA. [2] Department of Chemistry and Chemical Biology, Northeastern University, Boston, MA 02115, USA. [3] York Structural Biology Laboratory, Department of Chemistry, University of York, York YO10 5DD, UK. [4] Department of Bioengineering, Northeastern University, Boston, MA 02115, USA. [5] Present address: LAMBE, Université d'Evry Val d'Essonne, Université de Cergy Pontoise, CNRS, CEA, Université Paris-Saclay, Evry F-91025, France. These authors contributed equally: Benjamin Cressiot, Sandra J. Greive. Correspondence and requests for materials should be addressed to A.A.A. (email: fred.antson@york.ac.uk) or to M.W. (email: wanunu@neu.edu)

The advent of single-molecule detection is having an unparalleled impact on the speed with which structural and dynamic aspects of molecules can be probed[1]. In this regard, nanopores have shown much promise as electrical[2–7] and combined electro-optical sensors[8–10] and several nanopore-based systems are now being adopted as primary tools for DNA[11–13] and RNA[14] sequencing. Despite recent progress, identification and quantification of molecular species in solution[15–28] requires a reproducible nanopore platform that affords physical stability and structural precision. Sophisticated applications, such as electro-optical sensing, necessitate the pore position be geometrically defined to allow precise alignment of the optical system. While synthetic nanopores fabricated in solid-state (SS) membranes offer physical robustness[29–31], pore-to-pore variability often limits the reproducibility of experiments, necessitating additional control checks and validation. On the contrary, protein channels embedded in organic thin membrane, e.g., a lipid bilayer, offer the highest reproducibility due to the precise folding and repetitive nature of the constituting multisubunit protein oligomers[32,33], but their supporting membrane is typically less chemically and physically robust and, further, the pore position is variable due to in-plane diffusion of the protein channel[34]. While the use of amphiphilic polymers and polymerizable lipids have improved the lifetime, mechanical stability and voltage tolerance of the biomimetic support membranes for biological nanopores, as evidenced by the devices produced by Oxford Nanopore Technologies, SS membranes are likely to afford additional pressure and temperature resistance. Hybrid nanopore devices, in which channel-containing proteins are embedded in larger pores made in a SS matrix, have been proposed as a strategic solution for combining the benefits, while overcoming the limitations of existing nanopores[35]. Although initial experiments based on inserting pore-containing proteins with lipophilic regions into a SS pore looked promising[35], challenges in inserting such proteins into a SS pore and in controlling the protein orientation have remained major obstacles in the applicability of hybrid nanopores to nanotechnology.

Inspired by natural DNA pores, we designed a novel lipid-free hybrid nanopore based on the hydrophilic portal protein derived from the thermostable virus *G20c*[36]. In double-stranded DNA viruses, the portal protein is incorporated into the capsid shell (Fig. 1a), thereby serving as a natural pore through which DNA is moved in and out[37]. The protein contains a tight tunnel constriction with a repetitive chemical character, being made up by a circle of identical tunnel loops, contributed by 12 subunits[38]. In previous work, we engineered this protein to reprogram its physico-chemical properties (GG)[33] to create a portal with a larger minimum aperture of ~2.3 nm defined by the substitution of two bulky residues at the tip of the tunnel loop with glycines. Additional features of this portal system include substitution of an externally facing residue, located around the outside of the protein, by cysteine (designated C, e.g., CGG or CD/N) which allows chemical labeling and surface immobilization of the portal protein, as we have previously demonstrated for the lipid membrane insertion of the portal protein[33]. In this work, we have electrostatically engineered a portal protein variant (D/N) by replacing aspartic acid (D) residues at the internal tunnel surface with asparagines (N). This altered the charge of the lower part of the internal tunnel's surface from negative to positive (Fig. 1b and Supplementary Figure 1), a change that was crucial for electrical sensing of net negatively charged biomolecules. Here, we use this structurally programmable portal protein as a nanoscale adapter by electrokinetically embedding it snugly inside a larger pore made in a freestanding silicon nitride (SiN) membrane to form a lipid-free hybrid nanopore. Using the CD/N portal with engineered internal pore properties (Fig. 1b and Supplementary

Figure 1), we characterized the electrical properties of the hybrid pore and applied it to electrically detect different biomolecules. We demonstrate that a folded protein larger than the pore interior does not enter the hybrid portal. In contrast, biopolymers including single-stranded DNA (ssDNA), double-stranded DNA (dsDNA) that contains a single-stranded tail, and a peptide predicted to have a random coil conformation with a 10-amino acid α-helix at the C terminus can all be discriminated based on their distinct signal amplitudes in a way that is commensurate with their molecular cross-section. Our results indicate that the hybrid portal is a versatile sensor of various biopolymer types which may, with further development, find uses in genomic mapping as well as polypeptide and oligonucleotide sequencing.

## Results

**Portal is a nanoscale adapter for SS nanopores**. Electrokinetic corking of the *G20c* portal protein into the SS nanopore occurs when the force on the protein, induced by applied voltage, is sufficient to squeeze the portal into the SS pore (Figs. 1c, 2 and Supplementary Figures 2, 3). We find that stable insertion required specific geometric parameters for the SS nanopore, namely, a diameter between 5.4 to 6 nm and a nominal membrane thickness of 30 nm. Given the dimensions of the portal assembly[33] (Fig. 1b), the geometric constraints set by the SS pore restrict the range of possible orientations of the portal pore in it, such that the stem is inserted within the SS nanopore constriction, and the wider cap self-orients towards the top of the *trans* chamber (Fig. 1d). The larger size of the cap, as compared with the SS pore diameter, prevents the entire protein from moving through the SS nanopore. Remarkably, interactions between the portal protein squeezed into the SS pore and the SS pore surface contribute to a stable, self-inserting and self-aligning hybrid (Fig. 1d) that exhibits tolerable peripheral ion leakage, probed using cyclodextrin as a pore current modulator. Our hybrid pores exhibit lifetimes of hours (see Supplementary Figure 4), and similar ion current noise values to a lipid bilayer-supported portal protein nanopore[33] (Fig. 2 and Supplementary Figure 5).

After confirming the base current of stable SS nanopores of the desired diameter, addition of the portal protein to the *trans* chamber resulted in reversible partial blockades of the ionic current (Fig. 2a, b, Supplementary Figures 2 and 3). We interpret these short-lived events as portal protein collisions with the SS nanopore without stable insertion, where the ion current is partially blocked as the protein approaches the SS pore, prior to movement away. These short-lived events were usually followed by long-lived events (Fig. 2b, Supplementary Figures 2, 3), of comparable current blockade levels, events that were only observed in SS pores with diameters of 5.4 to 6 nm. We interpret the long-lived events as stable insertion of a portal protein into the SS nanopore to form a hybrid nanopore. The average conductance (mean and s.d.) of these hybrid pores was calculated (Fig. 2e) to be $1.50 \pm 0.48$ nS and $1.33 \pm 0.42$ nS for the CD/N (from 32 hybrid nanopores) and the CGG (from 15 hybrid nanopores) variants, respectively. Surprisingly, such hybrid pores remain stable at both positive and negative voltages (Fig. 2c). However, applying significant negative bias generally results in uncorking of the protein from the SS nanopore, and an example of an ejection at $-80$ mV is shown in Fig. 2c (red markers). These data are consistent with the protein insertion and removal being electrokinetically driven.

Obtaining a sufficient increase in the signal-to-noise ratio is a major challenge for properly identifying transport events by nanopore sensing. Power spectral densities of the current noise for a SS nanopore before, and after, insertion of a portal protein (Fig. 2d and Supplementary Figure 4) showed that the 1/f noise at

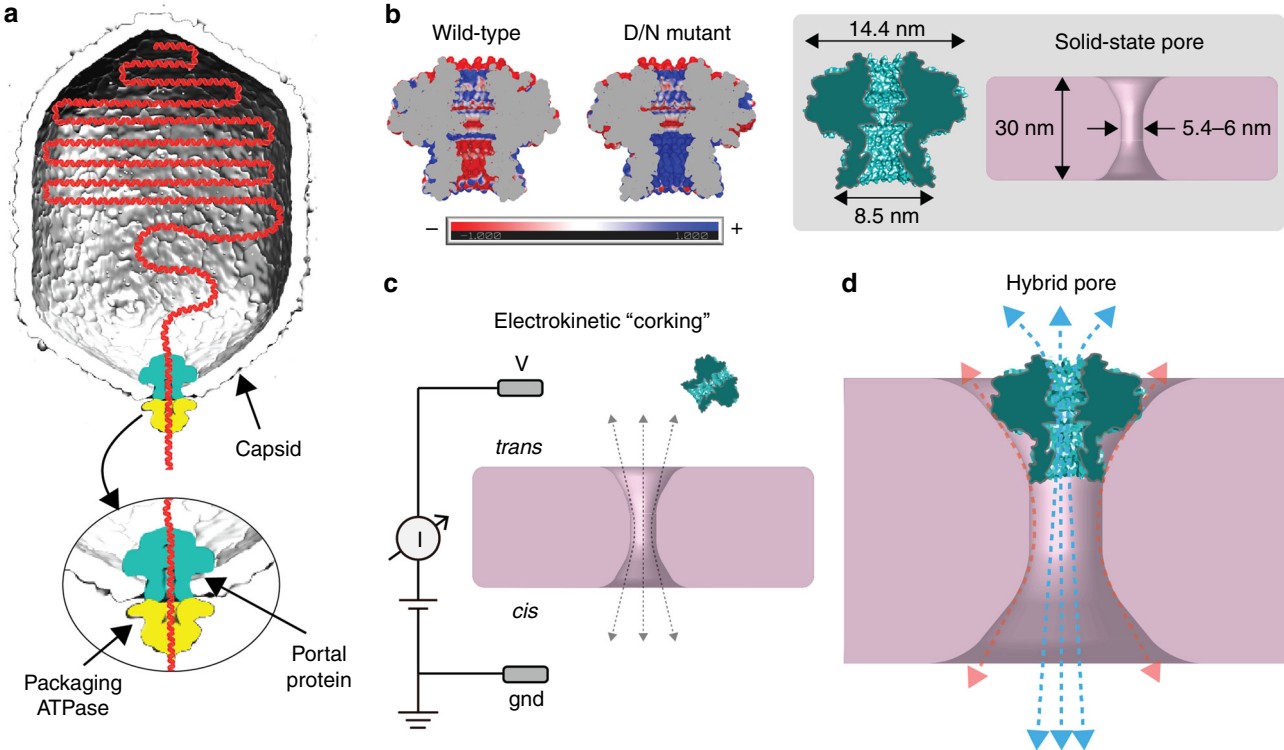

**Fig. 1** Design of a bio-inspired lipid-free hybrid nanopore. **a** Cartoon of the DNA packaging machine of a dsDNA virus. Viral genomic DNA (red) is translocated into the preformed virus capsid by the packaging ATPase (yellow) through the portal protein (aqua) embedded in viral capsid (gray). **b** Left, electrostatic properties of the tunnel in wild-type and mutant portal proteins. Slice through the middle of molecular surface colored according to charge from red ($-1\,kT\,e^{-1}$) to blue ($+1\,kT\,e^{-1}$). **b** Right (gray box), dimensions of the portal protein (teal, L) and the SS nanopore (pink, R). **c** Insertion of the purified portal protein into a nanopore fabricated in a thin solid-state (SS) membrane. Portal protein is applied to the *trans* chamber of a SS nanopore device containing an electrolyte solution of 20 mM Tris pH 7.5, 0.5 M NaCl. The protein electrokinetically inserts into the SS pore during application of a positive voltage. **d** Cartoon image of the hybrid pore, in which application of voltage results in ion current through the pore (blue arrows), as well as leakage current that is peripheral to the pore (red arrows)

low frequencies decreased upon formation of the hybrid pore. This 1/f noise reduction is consistent with a reduced pore conductance, as well as an indicator of the reduced surface charge fluctuations that are hallmarks of silicon nitride surfaces[39]. This, along with the observation that capacitance-dominated noise at high frequencies was comparable for both pores, suggests that no new source of noise was introduced by hybrid pore formation. We deduced that the observed variation in the open pore current for different hybrid nanopores (Supplementary Figures 2, 3) was likely to be caused by differences in SS geometry and the associated leakage currents around the portal protein. We attempted to measure the extent of ion leakage from the pore by measuring β-cyclodextrin interactions with the hybrid pore for the CGG mutant, a mutant that we previously embedded into a lipid membrane (Supplementary Figure 6)[33]. Our results show that β-cyclodextrin does not translocate the pore, in contrast to the same experiment conducted on the lipid-embedded version of the same portal protein. While this precludes an accurate measurement of the leakage, these results suggest that corking the protein into a snug SS nanopore slightly reduces the innermost pore constriction. The reproducible signals obtained from biomolecules, as well as the steady baselines of the hybrid, allow current blockades as low as ~20 pA to be accurately measured (Supplementary Figure 6). These data demonstrate that despite a low level of steady peripheral leakage, these hybrid pores are unique lipid-free protein-based pore sensors.

**Hybrid nanopores as sensors of biomolecules**. We then investigated the sensing capabilities of these hybrid nanopores by analyzing the transport of a peptide, comprising residues 1–43 of the human TPX2 protein, as a function of applied voltage (Fig. 3). The TPX2 peptide is negatively charged at pH 7.5 (pI = 3.7) and was added to the *cis* chamber, on the opposite side of the membrane to which the portal protein was introduced (see inset to Fig. 3b) to facilitate electrophoretically driven translocation. Adjusting the applied voltage from +30 to + 60 mV resulted in an increased baseline ion current through the hybrid pore, as well as the frequency of observed current blockades (Fig. 3a). Two kinds of current blockades associated with two different events were detected: bumping events, characterized by brief, low-level current blockades, arising from diffusion of the peptide close to the hybrid pore entrance; and translocation events, characterized by larger current blockades of longer duration. These two types of events are typically seen during translocation of DNA[40–42] and proteins[43–45] through protein channels. The inter-event time distribution is well fit by a single exponential equation (Supplementary Figure 7). The entry frequency (Fig. 3b) of the peptide into the hybrid pore is described by Van't Hoff Arrhenius relationship[44,46] $f = f_0\,\exp(V/V_0)$, consistent with both translocation of DNA[40–42], proteins[43,47] and peptides[46,48–51] through either α-hemolysin or aerolysin; and a significant entropic barrier for peptide entry into the pore. The dwell time distributions were well fit by a double-exponential equation (Supplementary Figure 7b), which are typically due to two types of processes,

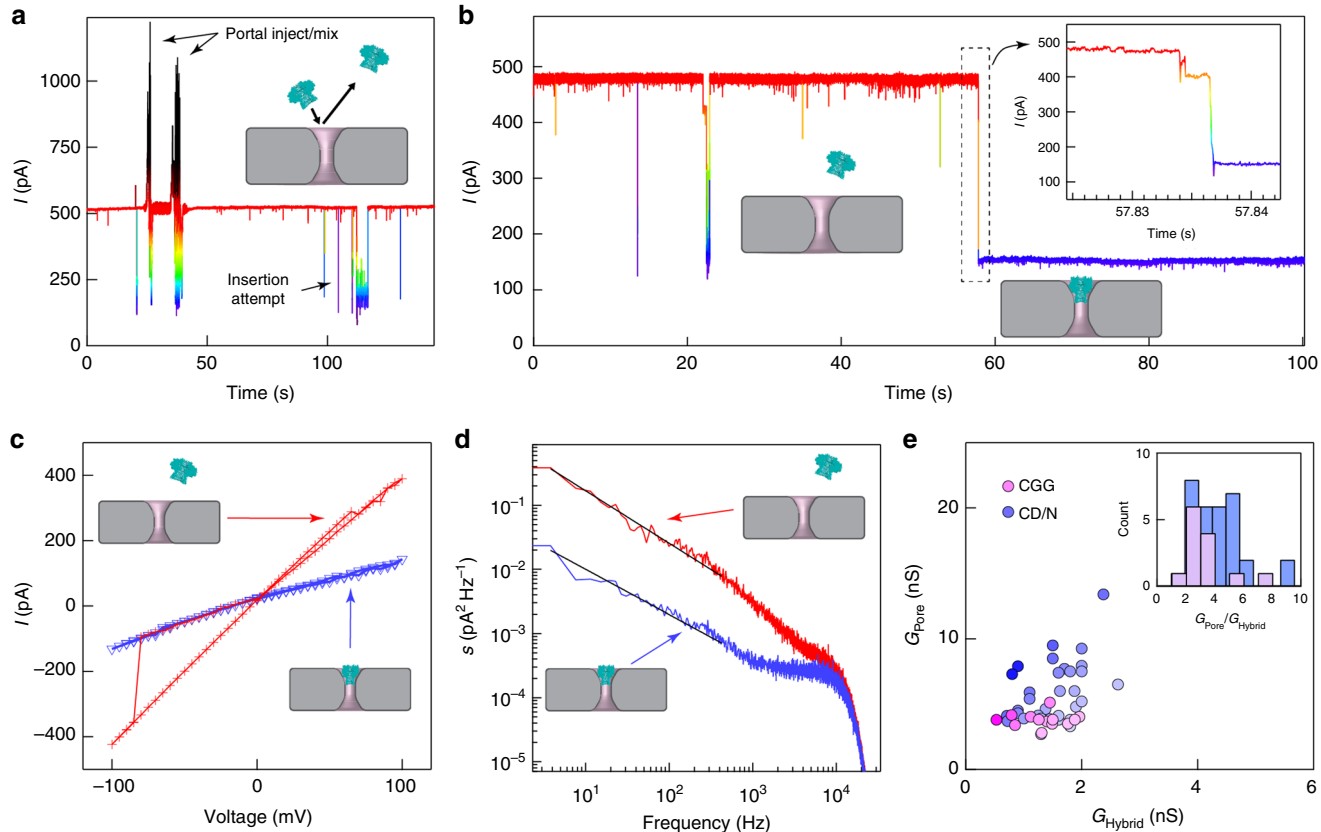

**Fig. 2** Characterization of the hybrid pore formation. **a** Typical current profile over time recorded through a 5.5 nm diameter SS pore at +100 mV. After injection of 0.1 nmol of portal protein, short current drops are detected, interpreted as portal collisions with the solid-state nanopore. **b** A representative current vs time trace recorded for a 5.4 nm SS nanopore at +80 mV, showing stable insertion of a portal protein. **c** Current as a function of the applied voltage for a 5.7 nm diameter SS pore recorded before (red markers) and after insertion of a portal protein (purple). **d** Current noise analysis of a 5.5 nm diameter solid-state nanopore before (red) and after insertion of a portal protein (purple). **e** Conductance of solid-state nanopore vs conductance of hybrid SS-portal pores formed after electrokinetic corking of the portal protein within the SS nanopore ($n = 32$ for CD/N hybrids and $n = 15$ for CGG hybrids). Experiments were performed in 0.5 M NaCl, 20 mM Tris, pH 7.5

normally associated with short bumping and longer translocation events[16]. We found that the average frequency for both types of events increases exponentially (Fig. 3b), while the average dwell time for the long events decreased exponentially with the applied voltage (Fig. 3c). Based on prior work that employed the α-hemolysin and aerolysin nanopores,[43,45] we conclude that the long events represent transport of the peptide through the hybrid pore to the *trans* chamber.

In order to further demonstrate the sensing capabilities of this hybrid pore, we investigated the transport of other biopolymers: dsDNA that contains a ssDNA tail, ssDNA, a folded protein as well as the TPX2 peptide (Fig. 4 and Supplementary Figures 7-10). Since all of these polymers are negatively charged at pH 7.5, electrophoresis allows molecular capture into the base of the portal protein following their introduction to the *cis* chamber (the opposite side of the SS membrane to portal insertion). After addition of each biopolymer: 36.0 μM insulin (Fig. 4a and Supplementary Figure 8), 7.7 μM hairpin-polydT$_{50}$ (Fig. 4b and Supplementary Figure 9), 10.3 μM TPX2 peptide (Figs. 3, 4c), 6.9 μM 60bp-polydT$_{30}$ (Supplementary Figure 10) and 16.6 μM ssDNA polydA$_{20}$dC$_{20}$dA$_{20}$ (Fig. 4d and Supplementary Figure 11), reversible partial blockades of the ionic current are observed at +60 mV. Similar short-lived bumping events as well as longer events were observed for each biopolymer, as described above for the TPX2 peptide (Fig. 3). These types of blockades were also observed at several different voltages for DNA molecules (Supplementary Figures 9–11), with voltage-dependent changes

in event frequency and duration for ssDNA polydA$_{20}$dC$_{20}$dA$_{20}$ consistent with translocation occurring (Supplementary Figure 11), as noted for the peptide above (Fig. 3). Conversely, the folded, globular molecule of insulin with a smallest dimension of ~3 nm (Protein Data Bank (PDB) code: 1zeh)[52,53] is too large for the ~2 nm constriction of the hybrid pore and therefore does not translocate. It is however possible that insulin explores the cavity at the portal tunnel's entrance (~5 nm) without being transported to the *trans* chamber, producing structured events that are long-lived and have a low current blockade level. Such events have been previously observed for nanoreactors, where biomolecules are captured or tethered within ClyA and FraC nanopores[20,54].

Lastly, we compare the event characteristics for different biopolymers at the same applied voltage of +60 mV by overlaying their scatter plots of ΔI vs. dwell time, as shown in Fig. 4e. Crucially, the level of current blockade, ΔI, appeared to be biopolymer dependent. We found current blockades (mean and s. d.) for the dsDNA (Fig. 4e) of $\Delta I = 34.6 \pm 4.2$ pA (5883 events), while in contrast, we found $\Delta I = 18.1 \pm 3.2$ pA (18,812 events) for ssDNA. This is nearly 2 times less than for the partially dsDNA, and is consistent with values found for dsDNA and ssDNA in SS nanopores[55], where the difference in conductance was found to be ~2.75-fold. For the peptide, we found $\Delta I = 30.1 \pm 5.5$ pA (3368 events). Since we showed that the peptide is transported through the pore (Fig. 3) and is predicted to contain an α-helix of ~1.4 nm in diameter, as seen in the structure of the TPX2 peptide bound to its partner kinase, Aurora A (PDB: 1ol5), the data are compatible

with translocation through the narrowest constriction of the hybrid pore (~2 nm diameter). The $\Delta I$ value found for the peptide is consistent with the α-helical region being the main cause of the blockade, and with its diameter being intermediate between that of dsDNA and ssDNA. These data suggest that the predicted α-helix is present in the isolated peptide under these experimental conditions. While transport of structured biopolymers has been reported for nucleic acids[56,57], only a single report to our knowledge presents transport of an α-helical peptide through a protein nanopore[45].

## Discussion

Hybrid nanopores, supported by SS membranes, could offer superior properties to both the planar lipid bilayer-based pores (that are sensitive to temperature, osmotic pressure and applied electric field strength and suffer from uncontrollable positional parameters) and SS nanopores (that are prone to edge erosion and are difficult to reproducibly fabricate with diameters <5 nm). However, despite having been the subject of industrial and academic research, development of a device that can be easily fabricated has proven difficult. For example, producing a hybrid pore

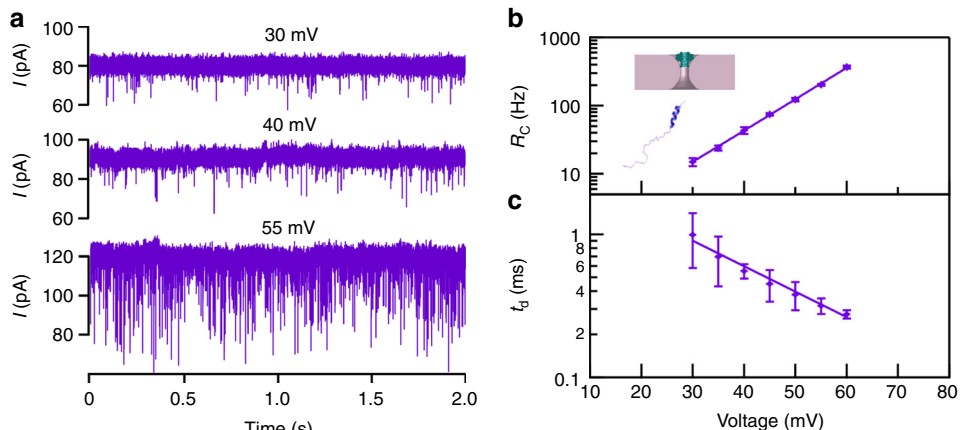

**Fig. 3** Dynamics of TPX2 peptide transport. **a** Current vs time trace recorded through a hybrid pore at +30, +40 and +55 mV in the presence of 10.3 µM TPX2 peptide. **b** Semi-log plot of the event frequency as a function of the applied voltage. **c** Semi-log plot of the peptide dwell time as a function of the applied voltage. The lines in (**b**, **c**) are exponential fits to the data. Experiments were performed in 0.5 M NaCl, 20 mM Tris, pH 7.5. Data shown in semi-log plots are mean and s.d. of 23,516 events (total for all points) from one hybrid nanopore

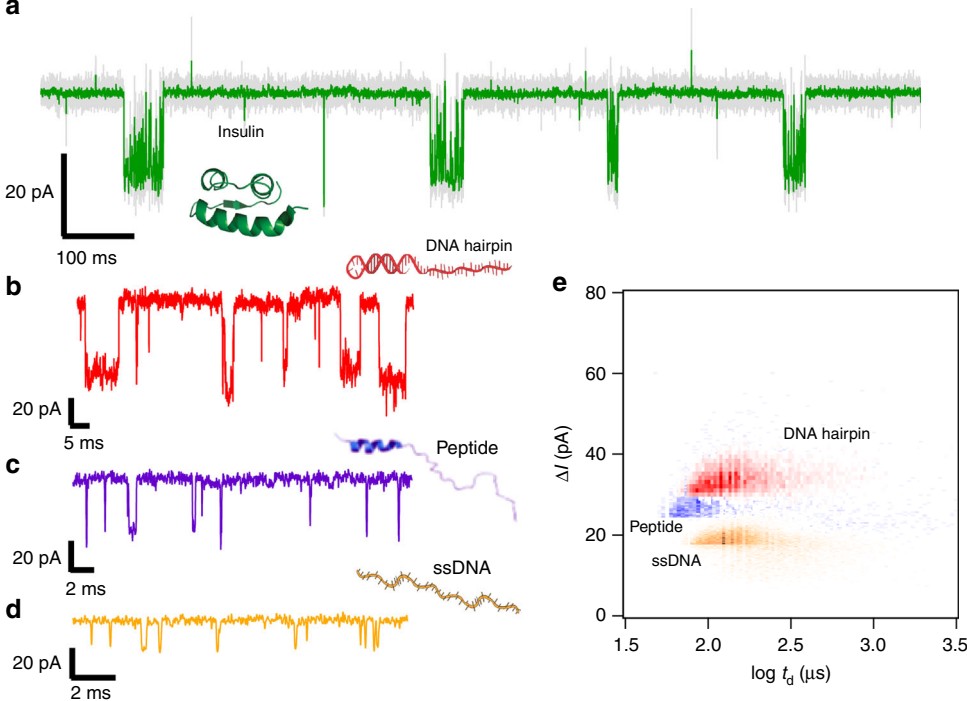

**Fig. 4** Sensing different biopolymers using a hybrid nanopore. Current vs time trace recorded through the hybrid pore at +60 mV in the presence of **a** 36.0 µM insulin, **b** 7.7 µM DNA hairpin, **c** 10.3 µM TPX2 peptide and **d** 16.6 µM ssDNA. The data in **a** were filtered at 10 kHz (gray) or 0.5 kHz (green). **e** Scatter plot of $\Delta I$ vs dwell time for the DNA hairpin (red, $n = 5883$ events), the peptide (purple, $n = 3368$ events) and the ssDNA (orange, $n = 18,812$ events)). Experiments were performed in 0.5 M NaCl, 20 mM Tris, pH 7.5

based on DNA origami or α-hemolysin, a membrane protein, noted substantial leakage current[58,59] or relatively short hybrid pore lifetime and required complex protein modifications[35]. In contrast, the hybrid nanopore described here is based on a soluble, stable and relatively hydrophilic viral portal protein, whose chemical properties, including those inside tunnel, can be easily tuned. This work demonstrates that the lipid-free hybrid nanopore comprising the *G20c* portal protein inserted into a thin SS SiN membrane is easy to assemble, with the portal protein readily electrokinetically inserting into the SS pores and typically remaining stable for hours of experimental time. We plainly demonstrate the utility of this hybrid pore as a nanosensor, by observing characteristic readout for dsDNA and ssDNA, as well as a peptide and a globular protein.

The stability and tunability of the hybrid nanopore that combines the advantages of both protein and SS nanopores encourages further work to introduce specific sensing properties for this hybrid pore and produce robust and geometrically controlled devices amenable to sophisticated high-throughput array-based or coupled multi-system (e.g., optical, flow or pressure) applications in nanotechnology.

## Methods

**Protein engineering and purification**. Mutant portal proteins CGG[33] and CD/N (Supplementary Figure 1) were expressed in *Escherichia coli* Shuffle cells at 30 °C overnight after induction with 0.5 mM isopropyl β-D-1-thiogalactopyranoside (IPTG) at 0.8 $OD_{600}$. The cells were lysed by sonication on ice in 50 mM Tris pH 8, 1 M NaCl, 10 mM imidazole, 100 mM AEBSF, 10 mg mL$^{-1}$ lysozyme and 2 mM dithiothreitol (DTT). After clarification by centrifugation at 15,000 rpm for 15 min, the protein was purified from the lysate by Immobilized Metal Affinity Chromatography (IMAC; 5 mL HiTrap FF Crude, GE Healthcare) and eluted over a gradient of 10–500 mM imidazole over 10 column volumes. Fractions containing the protein were subjected to a buffer exchange step over a desalting column (HiPrep 26/10; GE Healthcare) to improve 3C cleavage of the histidine affinity tag (50 mM Tris pH 8, 0.5 M NaCl, 50 mM K Glutamate, 1 mM DTT) prior to buffer exchange back into low imidazole buffer before the second IMAC step and purification to homogeneity in 20 mM Tris pH 8, 1 M NaCl, 1 mM DTT, before freezing in liquid nitrogen and storage at −80 °C. Protein was exchanged into 20 mM Tris pH 7.5, 0.5 M NaCl buffer (Zeba Spin Columns, Thermofisher) for use in hybrid nanopore formation. CD/N mutant proteins were characterized for stability and assembly state by nano differential scanning fluorimetry (nanoDSF) and negative stained transmission electron microscopy (TEM; Supplementary Figure 1). Human TPX2$_{1-43}$ peptide was produced in *E. coli* as a his-tagged GB1 fusion (Marko Hyvonen, University of Cambridge). Expression was induced in BL21 pLysS cells at 0.8 $OD_{600}$ with 0.5 mM IPTG for 4 h at 37 °C. The fusion protein was purified by IMAC in standard nickel affinity chromatography buffers at pH 7.5 containing 1 mM DTT and eluted over a 10–500 mM imidazole gradient, after which peak fractions were pooled, prior to buffer exchange (desalting column, GE Healthcare) into 3C cleavage buffer 50 mM Tris pH 7.5, 0.5 M NaCl, 10 mM imidazole and 1 mM DTT. After cleavage, a second IMAC step removed the his-tagged GB1 fragment and the unbound fraction containing the TPX2 peptide (GPGSMLSYSY-DAPSDFINFSSLDDEGDTQNIDWFEEKANLENLKGGGCQ) was concentrated by centrifugal ultrafiltration using a 3 kDa cutoff filter (Amicon) and further purified over a S75 10/300 size exclusion column (GE Healthcare) in 20 mM Tris pH 7.5, 0.5 M NaCl and 1 mM DTT prior to concentration as before, freezing in liquid nitrogen and storage at −80 °C.

**Experimental set-up**. Nanopores were fabricated in 30 nm thick SiN membranes using previously reported methods[60,61]. The pore diameters ranged between 5.4 and 6 nm in order to properly orient the portal protein. Nanopores were cleaned with hot piranha (3:1 $H_2SO_4$/ $H_2O_2$), followed by hot deionized water before each experiment. After being dried under vacuum, nanopore chips were assembled in a custom cell equipped with Ag/AgCl electrodes, and quick-curing silicone elastomer was applied between the chip and the cell to seal the device and thereby reduce the noise by minimizing the chip capacitance. We introduced 0.5 M NaCl, 20 mM Tris, pH 7.5, as an electrolyte solution onto both sides of the chip. Portal protein was always added to the *trans* chamber and the biopolymers to the *cis* chamber. All experiments were carried out at ambient temperature. Human insulin was purchased from Alfa Aesar (Thermofisher), and dsDNA hairpin (5′-GCTGTCT GTTGCTCTCTCGCAACAGACAGC T$_{50}$-3′), ssDNA (5′-dA$_{20}$dC$_{20}$dA$_{20}$-3′), 60 bp-polydT30 ((5′-TCAGGGTTTTTTTACT)$_4$ T$_{30}$-3′) and its complementary strand ((3′-AGTAAAAAAACCCTGA-5′)$_4$) were synthesized by Integrated DNA Technology.

**Electrical detection and data acquisition**. The ionic current through SS nanopores and portal hybrid protein was measured using an Axopatch 200B amplifier (Molecular Devices). Data were filtered at 10 kHz and acquired at 250 kHz using a National Instruments DAQ card and custom LabVIEW software. Data were processed and events were detected using Python software (https://github.com/rhenley/Pyth-Ion/). The values for the open pore current ($I_0$) and the standard deviation of the noise ($σ$) were extracted from the analysis. The threshold (Th) applied in Python to separate events from the noise is given by Th = $I_0 - 4σ$. The average duration of blockades was deduced from the distribution of blockade duration, $τt$. The two blockade time distributions of independent events are adjusted with a double-exponential function, $y = A_1\exp(t/τ_1) + A_2\exp(t/τ_2)$. All statistical analyses were performed using Igor Pro software (WaveMetrics Inc.).

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

# ARTICLE

24. Baaken, G. et al. High-resolution size-discrimination of single nonionic synthetic polymers with a highly charged biological nanopore. *ACS Nano* **9**, 6443–6449 (2015).
25. Fennouri, A. A. et al. Single molecule detection of glycosaminoglycan hyaluronic acid oligosaccharides and depolymerization enzyme activity using a protein nanopore. *ACS Nano* **6**, 9672–9678 (2012).
26. Lee, J. et al. Semisynthetic nanoreactor for reversible single-molecule covalent chemistry. *ACS Nano* **10**, 8843–8850 (2016).
27. Willems, K., Van Meervelt, V., Wloka, C. & Maglia, G. Single-molecule nanopore enzymology. *Philos. Trans. R. Soc. Lond. B Biol. Sci.* **372**, pii: 20160230 (2017).
28. Rosen, C. B., Rodriguez-Larrea, D. & Bayley, H. Single-molecule site-specific detection of protein phosphorylation with a nanopore. *Nat. Biotechnol.* **32**, 179–181 (2014).
29. Verschueren, D. V., Jonsson, M. P. & Dekker, C. Temperature dependence of DNA translocations through solid-state nanopores. *Nanotechnology* **26**, 234004 (2015).
30. Oukhaled, A. et al. Dynamics of completely unfolded and native proteins through solid-state nanopores as a function of electric driving force. *ACS Nano* **5**, 3628–3638 (2011).
31. Yamazaki, H. et al. Label-free single-molecule thermoscopy using a laser-heated nanopore. *Nano. Lett.* **17**, 7067–7074 (2017).
32. Song, L. et al. Structure of staphylococcal alpha-hemolysin, a heptameric transmembrane pore. *Science* **274**, 1859–1866 (1996).
33. Cressiot, B. et al. Porphyrin-assisted docking of a thermophage portal protein into lipid bilayers: nanopore engineering and characterization. *ACS Nano* **11**, 11931–11945 (2017).
34. Castell, O. K., Berridge, J. & Wallace, M. I. Quantification of membrane protein inhibition by optical ion flux in a droplet interface bilayer array. *Angew. Chem. Int. Ed.* **51**, 3134–3138 (2012).
35. Hall, A. R. et al. Hybrid pore formation by directed insertion of α-haemolysin into solid-state nanopores. *Nat. Nanotech.* **5**, 874–877 (2010).
36. Williams, L. S., Levdikov, V. M., Minakhin, L., Severinov, K. & Antson, A. A. 12-Fold symmetry of the putative portal protein from the Thermus thermophilus bacteriophage G20C determined by X-ray analysis. *Acta Crystallogr. Sect. F Struct. Biol. Cryst. Commun.* **69**, 1239–1241 (2013).
37. Casjens, S. R. & Gilcrease, E. B. Determining DNA packaging strategy by analysis of the termini of the chromosomes in tailed-bacteriophage virions. *Methods Mol. Biol.* **502**, 91–111 (2009).
38. Lebedev, A. A. et al. Structural framework for DNA translocation via the viral portal protein. *EMBO J.* **26**, 1984–1994 (2007).
39. Hoogerheide, D. P., Garaj, S. & Golovchenko, J. A. Probing surface charge fluctuations with solid-state nanopores. *Phys. Rev. Lett.* **102**, 256804 (2009).
40. Henrickson, S. E., Misakian, M., Robertson, B. & Kasianowicz, J. J. Driven DNA transport into an asymmetric nanometer-scale pore. *Phys. Rev. Lett.* **85**, 3057–3060 (2000).
41. Meller, A. & Branton, D. Single molecule measurements of DNA transport through a nanopore. *Electrophoresis* **23**, 2583–2591 (2002).
42. Japrung, D., Henricus, M., Li, Q., Maglia, G. & Bayley, H. Urea facilitates the translocation of single-stranded DNA and RNA through the α-hemolysin nanopore. *Biophys. J.* **98**, 1856–1863 (2010).
43. Cressiot, B. et al. Dynamics and energy contributions for transport of unfolded pertactin through a protein nanopore. *ACS Nano* **9**, 9050–9061 (2015).
44. Pastoriza-Gallego, M. et al. Dynamics of unfolded protein transport through an aerolysin pore. *J. Am. Chem. Soc.* **133**, 2923–2931 (2011).
45. Oukhaled, A., Bacri, L., Pastoriza-Gallego, M., Betton, J.-M. & Pelta, J. Sensing proteins through nanopores: fundamental to applications. *ACS Chem. Biol.* **7**, 1935–1949 (2012).
46. Stefureac, R., Long, Y.-T., Kraatz, H.-B., Howard, P. & Lee, J. S. Transport of alpha-helical peptides through alpha-hemolysin and aerolysin pores. *Biochemistry* **45**, 9172–9179 (2006).
47. Pastoriza-Gallego, M. et al. Evidence of unfolded protein translocation through a protein nanopore. *ACS Nano* **8**, 11350–11360 (2014).
48. Wang, H.-Y., Ying, Y.-L., Li, Y., Kraatz, H.-B. & Long, Y.-T. Nanopore analysis of β-amyloid peptide aggregation transition induced by small molecules. *Anal. Chem.* **83**, 1746–1752 (2011).
49. Sutherland, T. C. et al. Structure of peptides investigated by nanopore analysis. *Nano Lett.* **4**, 1273–1277 (2004).
50. Meng, H. et al. Nanopore analysis of tethered peptides. *J. Pept. Sci.* **16**, 701–708 (2010).
51. Mereuta, L. et al. Slowing down single-molecule trafficking through a protein nanopore reveals intermediates for peptide translocation. *Sci. Rep.* **4**, 3885–3885 (2014).
52. Whittingham, J. L., Edwards, D. J., Antson, A. A., Clarkson, J. M. & Dodson, G. G. Interactions of phenol and m-cresol in the insulin hexamer, and their effect on the association properties of B28 pro -->Asp insulin analogues. *Biochemistry* **37**, 11516–11523 (1998).
53. Kadima, W. et al. The influence of ionic strength and pH on the aggregation properties of zinc-free insulin studied by static and dynamic laser light scattering. *Biopolymers* **33**, 1643–1657 (1993).
54. Van Meervelt, V. et al. Real-time conformational changes and controlled orientation of native proteins inside a protein nanoreactor. *J. Am. Chem. Soc.* **139**, 18640–18646 (2017).
55. Skinner, G. M., van den Hout, M., Broekmans, O., Dekker, C. & Dekker, N. H. Distinguishing single- and double-stranded nucleic acid molecules using solid-state nanopores. *Nano Lett.* **9**, 2953–2960 (2009).
56. Lin, J., Fabian, M., Sonenberg, N. & Meller, A. Nanopore detachment kinetics of poly(A) binding proteins from RNA molecules reveals the critical role of C-terminus interactions. *Biophys. J.* **102**, 1427–1434 (2012).
57. Akeson, M., Branton, D., Kasianowicz, J. J., Brandin, E. & Deamer, D. W. Microsecond time-scale discrimination among polycytidylic acid, polyadenylic acid, and polyuridylic acid as homopolymers or as segments within single RNA molecules. *Biophys. J.* **77**, 3227–3233 (1999).
58. Bell, N. A. W. et al. DNA origami nanopores. *Nano Lett.* **12**, 512–517 (2012).
59. Wei, R., Martin, T. G., Rant, U. & Dietz, H. DNA origami gatekeepers for solid-state nanopores. *Angew. Chem. Int. Ed.* **51**, 4864–4867 (2012).
60. Larkin, J. et al. Slow DNA transport through nanopores in hafnium oxide membranes. *ACS Nano* **7**, 10121–10128 (2013).
61. Wanunu, M. et al. Rapid electronic detection of probe-specific microRNAs using thin nanopore sensors. *Nat. Nanotech.* **5**, 807–814 (2010).

## Acknowledgements

This work was supported by a Bilateral United Kingdom Biotechnology and Biological Sciences Research Council (BBSRC)–United States National Science Foundation (NSF) Lead Agency Pilot Program grant (BB/N018729/1 and NSF-1645671 to A.A.A. and M.W., respectively). We would like to thank Marko Hyvonen, Department of Bio-chemistry, University of Cambridge (UK), for the expression plasmid for the TPX2 peptide. NanoDSF and negative stained TEM of mutant portal proteins were carried out in the Technology Facility, Department of Biology, University of York (UK).

## Author contributions

A.A.A., M.W. and S.J.G. conceived the study. B.C. and S.J.G. designed the engineered portal protein and the electrical sensing experiments. S.J.G. engineered, purified and biophysically characterized the portal protein variants. B.C. performed the electrical sensing experiments with DNA and protein, and analyzed the data with assistance from M.W. M.M. conducted the cyclodextrin electrical sensing experiments and analyzed the data under supervision of M.W. B.C., M.W. and S.J.G co-wrote the paper with assistance from A.A.A. All authors discussed the results and commented on the manuscript.

## Additional information

**Competing interests:** The authors declare no competing interests.

