## [Peer Review File · Nature Communications]

Reviewers' comments:

Reviewer #1 (Remarks to the Author):

Nanopore sensing is a rapidly developing field using the confinement in a nanometer sized channels to sense molecules with its most prominent application of DNA sequencing. In a previous work the authors used a capsid entry pore, functionalized the side chain and inserted this pore into a lipid bilayer (ACSnano, ref 33). Here in this work they extended their approach and inserted the capsid pore (portal protein) into a solid state pore. Although this is not the first solution of a combined "solid state corked with a biopore" this approach is novel.

This particular solution is interesting

- as the channel is the natural entry of DNA into the virus capsid so one may expect that this pore is optimized for DNA translocation. Moreover the pore seems to be large allowing further functionalization of the constriction zone.

and furthermore

- the solid state nanopore might provide a superior stability. Maybe in the future such an array can be produced in an arrays format allowing microstructuring.

Here the authors use the hybrid approach and show for two mutants the ion current spectra for several molecules to be detected: cyclodextrin, TPX2, DNA hairpin, or ssDNS.

This approach (although not the first) is novel and interesting for the field. In particular using the portal protein as sensing element with the possibility for further engineering might become be a new platform.

Below some suggestions:

- The authors should distinguish better the novelty of this ms with respect to their previous one.
- the general audience will have difficulties to understand the motivation and the impact of the different mutants. A few more sentences would help.

S5: the driving force for cyclodextrin is electroosmosis?

- the analysis of the TPX2 is not clear to me. The authors should give more details on their conclusions. What happens if TPX2 is added on the opposite side?

- is there more evidence that insulin is not permeating? What would be the minimal number of successful translocation to be detected?

I recommend publication

Reviewer #2 (Remarks to the Author):

Cressiot et al. present a very interesting article on designing a hybrid based nanopores which is lipid free. As the authors mention this is not the first time a hybrid pore has been designed and incorporated in a SS platform. However, they build and improve on this strategy. The method is novel and worthy of publication. I have the following comments and recommendations:

1. Abstract: Reference to solid-state pores being drilled in thin metallic membranes. This is not correct for either electrical or optical based detection. Drilling is not the only mechanism for SS pores and for optical sensing there are examples where non-metallic pores are used.

2. There is an emphasis in electro-optical sensing in abstract and intro. Not clear why this is and is rather confusing. Needs either clarifying or changing especially in the abstract.

3. In the intro, there is an argument that protein channels are not stable. However, this problem to some extent has been solved as the authors are aware (e.g. oxford nanopore). A clearer justification is needed as to the particular problem being solved and most certainly has to be put in appropriate context.

4. Recently there have been a few examples of hybrid nanopores using DNA origami – what are the advantages/disadvantages?
5. Number of events are often low and would benefit from more experimental data – e.g. figure 2e.
6. Not clear what the number of events is for each experiment.
7. Fig S7: some peaks appear biphasic- what is the cause?
8. According to the methods section, data was filtered at 10 kHz; however, events often appear quicker than this which will result in artefacts in the dwell time and amplitude. This should be clarified or alternatively reanalysed.

Reviewer #3 (Remarks to the Author):

The paper reports on the novel lipid-free hybrid nanopore-based on a natural DNA pore from a thermostable virus, electrokinetically inserted into a larger nanopore supported in a silicon nitride membrane. The authors present a carefully performed set of experiments, which are used to discuss the engineering and sensing properties of the hybrid nanopore. The information provided allows for replicating the experiments by other authors.

Unlike other hybrid pore system, authors claim to improve on the lifetime of the pores compared to the state of the art of hybrid nanopores (referring here to Hall et.al. 2010 Nature Nanotechnology) The robust geometry and improved noise performance of the hybrid system enabled sensing of several biomolecules TX2 peptide, ssDNA, harpin DNA and insulin.

I want to pinpoint a few remarks that are of importance to me and that the authors could help clarify.

1. The authors indicate at line 72 : “Our hybrid pores exhibit lifetimes of hours, and similar ion current noise values to a lipid bilayer supported portal protein nanopore³³ (Figure 2).” However, the trace is only 100 s long To support a lifetime of hours for hybrid pores it would be beneficial to prepare separate figure that shows a baseline of the same pore at the start of the experiment and at the end (several hours later).
2. As discussed in the paper biological pores, due to their stable and reproducible geometry, have stable and reproducible current drops. Authors argue that their hybrid nanopore preserves this advantage of the biological pores. However, nanopores are corks to the ss .pores of different diameters (SI Figure 2 and 3). Diameters of ss-pores range from 5.9-5.4 nm, consequently as reported by the authors the level of leakage current, and noise were correlated with the size of the ss-pore. Authors should comment more on how crucial is the size of ss-pore and suggest the strategy to compare the results from different hybrid pores. Ideally, one would like to take advantage and in the future perform high-throughput experiments. In the paper the authors should discuss how practical their approach would be in terms of high-throughput measurements.
3. SI. Figure 2 gives examples of portal pore CD/N insertions into the ss-nanopore for three different pore sizes, it is difficult to judge between the panels, but it looks like that the noise level is higher for the 5.5nm large ss-pore compared to panel c) ss-pore of 5.6 nm. It is also unclear what is a difference between panels b and c) pore is 5.6 nm large and bias conditions are the same, but the noise level appears larger in b). Is this an identical pore, what changed during the experiment...
4. Due to the tremendous progress of ss-nanopore fabrication techniques based on dielectric breakdown and light assisted nanopore drilling, I was wondering if authors attempted the to cork CGG into ss-pores fabricated in that way? How crucial is TEM since it appears as the preferred fabrication technique (ref. 58 and 59)? For supporting information IVs and TEM images of the ss-pores would be beneficial as the small changes in the pore size have a significant impact on the hybrid performance. –judging from the SI Figure 2.

Thermostable Virus Portal Proteins as Reprogrammable Adapters for Solid-State Nanopore Sensors

Red text: our response to each of the reviewers' comments.

Italicized and 'quoted' red text: changes we have made to the manuscript text.

Reviewer #1 (Remarks to the Author):

Recommendation: 'I recommend publication'

Comments: Nanopore sensing is a rapidly developing field using the confinement in a nanometer sized channels to sense molecules with its most prominent application of DNA sequencing. In a previous work the authors used a capsid entry pore, functionalized the side chain and inserted this pore into a lipid bilayer (ACS Nano, ref 33). Here in this work they extended their approach and inserted the capsid pore (portal protein) into a solid state pore. Although this is not the first solution of a combined "solid state corked with a biopore" this approach is novel.

This particular solution is interesting

- as the channel is the natural entry of DNA into the virus capsid so one may expect that this pore is optimized for DNA translocation. Moreover the pore seems to be large allowing further functionalization of the constriction zone.

and furthermore

- the solid state nanopore might provide a superior stability. Maybe in the future such an array can be produced in an arrays format allowing microstructuring.

Here the authors use the hybrid approach and show for two mutants the ion current spectra for several molecules to be detected: cyclodextrin, TPX2, DNA hairpin, or ssDNS.

This approach (although not the first) is novel and interesting for the field. In particular using the portal protein as sensing element with the possibility for further engineering might become be a new platform.

Below some suggestions:

- The authors should distinguish better the novelty of this ms with respect to their previous one.

We thank the reviewer for this comment. We now realize that the use of the term 'lipid-free' in the abstract did not adequately delineate the novelty of this work from the previous publication. Consequently, we have added the following textual changes to the introduction as follows:

'Inspired by natural DNA pores, we designed a novel *lipid-free* hybrid nanopore based on the hydrophilic portal protein derived from the thermostable virus G20c.'

'In previous work, we engineered this protein to reprogram its physicochemical properties (GG),³³ to create a portal with a larger minimum aperture of ~2.3 nm defined by the substitution of two bulky residues at the tip of the tunnel loop with glycines. Additional features of this portal system include substitution of an externally facing residue, located around the outside of the protein, by cysteine (designated "C" e.g. CGG or CD/N) which allows chemical labeling and surface immobilization, as we previously demonstrated for the lipid membrane insertion of the portal protein³³. In this work, we have electrostatically engineered a portal protein variant (D/N) by replacing aspartic acid (D) residues at the internal tunnel surface with asparagines (N). This altered the charge of the lower part of the internal tunnel's surface from negative to positive (Figure 1b and Supplementary Figure S1), a change that was crucial for electrical sensing of net negatively charged biomolecules.

Here, we use this structurally programmable portal protein as a nanoscale adapter by electrokinetically embedding it snugly inside a larger pore made in a freestanding silicon nitride (SiN) membrane, to form a lipid-free hybrid nanopore (Figures 1c, 2 & Supplementary Figures S2 & S3).'

- the general audience will have difficulties to understand the motivation and the impact of the different mutants. A few more sentences would help.

We appreciate the comment, and have clarified the design of the different mutant proteins, as described in the previous response.

S5: the driving force for cyclodextrin is electroosmosis?

Indeed, the driving force for the interaction of cyclodextrin with a CGG portal protein is an electro-osmotic flux (EOF), as demonstrated previously through the same pore at the same pH (Cressiot et al, ACS Nano, 2017). This observation can be attributed to EOF through the pore governed by K⁺ flow, in good agreement with the negative internal charges within the pore and extensive negative isocontours of the cap.

Note that this figure is now been altered to Figure S6 due to the insertion of an additional supplemental figure in response to a comment by reviewer 3, below. We added the following text to the legend for figure S6:

"The β -cyclodextrin molecules are driven electro-osmotically to the CGG portal protein due to a K⁺ flow, in good agreement with the negative internal charges within the pore and extensive negative isocontours, as previously demonstrated (ref)".

- the analysis of the TPX2 is not clear to me. The authors should give more details on their conclusions. What happens if TPX2 is added on the opposite side?

As described above, the changes to the internal charge of the tunnel introduced by the D/N mutants allowed for the sensing of biomolecules that have a net negative charge and are likely electrophoretically driven to the pore.

Since our corking scheme requires a positive bias, we typically add negatively charged molecules (TPX2 is negative at the pH we used) to the cis chamber, opposite to the side where the portal protein is added. Applying negative bias for prolonged period results in uncorking of the protein from the SS nanopore, as shown in Figure 2.

An additional consideration, as noted in our previous work (Cressiot et al, ACS Nano, 2017) and by the Guo laboratory for other portal proteins, the increased flexibility of the tunnel loops within the central channel along the tunnel axis in one direction (in this setup, towards the trans chamber) compared to the other, confers a directional bias to the translocation of biomolecules through this protein nanopore.

In order to clarify this point we have made the following change to the relevant text in the manuscript:

‘The TPX2 peptide is negatively charged at pH 7.5 ($pI = 3.7$) and was added to the cis chamber, on the opposite side of the membrane to which the portal protein was introduced (see inset to Figure 3b), *to facilitate electrophoretically driven translocation.*’

- is there more evidence that insulin is not permeating? What would be the minimal number of successful translocation to be detected?

We are guided in this assumption by two pieces of evidence: 1) The dimensions of insulin are simply too large to be compatible with translocation through the narrowest part of the portal protein, and 2) our observation of long-lived blockades, similar to those seen for protein entry into the large chamber of ClyA (Biesemans et. al. 2015; Soskine et. al. 2013), suggest prolonged interactions with the pore, which, given its size, suggest trapping of insulin inside the pore.

Reviewer #2 (Remarks to the Author):

Recommendation: ‘The method is novel and worthy of publication.’

Comments: Cressiot et al. present a very interesting article on designing a hybrid based nanopores which is lipid free. As the authors mention this is not the first time a hybrid pore has been designed and incorporated in a SS platform. However, they build and improve on this strategy. The method is novel and worthy of publication.

Cressiot et al. present a very interesting article on designing a hybrid based nanopores which is lipid free. As the authors mention this is not the first time a hybrid pore has been designed and incorporated in a SS platform. However, they build and improve on this strategy. The method is novel and worthy of publication.

I have the following comments and recommendations:

1. Abstract: Reference to solid-state pores being drilled in thin metallic membranes. This is not correct for either electrical or optical based detection. Drilling is not the only mechanism for SS pores and for optical sensing there are examples where non-metallic pores are used.

We agree and have replaced 'drilled' with 'fabricated' in the text as follows:

Abstract – ...' or solid-state (SS) pores *fabricated in thin metallic membranes.*'

Figure 1 legend – 'c) Insertion of the purified portal protein into a nanopore *fabricated in a thin solid-state (SS) membrane.*'

2. There is an emphasis in electro-optical sensing in abstract and intro. Not clear why this is and is rather confusing. Needs either clarifying or changing especially in the abstract.

We agree and have removed the reference to electro-optical in the abstract and clarified the relevant section in the introduction as follows:

Abstract – '**Current electrical sensing devices**'

Introduction – ...'and *combined* electro-optical sensors '

...'*requires a reproducible nanopore platform that affords physical stability and structural precision. Sophisticated applications, such as electro-optical sensing, necessitate the pore position be geometrically defined to allow precise alignment of the optical system.*

3. In the intro, there is an argument that protein channels are not stable. However, this problem to some extent has been solved as the authors are aware (e.g. oxford nanopore). A clearer justification is needed as to the particular problem being solved and most certainly has to be put in appropriate context.

This is a good point, consequently we have an additional sentence in the introduction:
'While the use of amphiphilic polymers and polymerizable lipids have improved the lifetime, mechanical stability and voltage tolerance of the biomimetic support membranes for biological nanopores, as evidenced the devices produced by Oxford Nanopore Technologies, SS membranes are likely to afford additional pressure and temperature resistance.'

4. Recently there have been a few examples of hybrid nanopores using DNA origami – what are the advantages disadvantages?

In fact, DNA origami has been used for the formation of hybrid nanopores (Bell et al, Nano Lett, 2012; Wei et al, Angew Chem, 2012; Ketterer et al, Nat Com, 2018) since it enables the construction of 3D shapes with nanoscale level accuracy in geometry and surface properties. Like recombinant proteins, identical copies are created in parallel since the method relies on self-assembly and can be applied to high throughput measurements. However, the studies show that there is a substantial leakage current through the structure and its seal, which

limits signals for translocating molecules through the hybrid nanopore. Also, there can be a significant increase in ionic current noise after formation of the hybrid nanopore, and we have observed that highly variable currents that result from different DNA origami trappings due to fluctuation of the origami on the solid-state nanopore are significant.

We have noted this with the addition of the following text in the discussion.

'For example, producing a hybrid pore based on *DNA origami* or α -hemolysin, a membrane protein, noted *substantial leakage currents*^{58,59} or relatively short hybrid pore lifetime and required complex protein modifications³⁵, *respectively*.'

5. Number of events are often low and would benefit from more experimental data – e.g. figure 2e.

We firmly believe that the number of experiments we have done is enough for statistical analysis: Figure 2e does not show a scatter plot of molecular sensing events, but the formation of 47 hybrid nanopores (32 CD/N and 15 CGG hybrid pores). The previous and only paper describing the formation of a hybrid pore using a protein inserted into a solid-state nanopore (Hall et al, Nature Nanotech, 2012) showed the formation of a total of 21 hybrid pores. Furthermore, the 47 hybrid nanopores we describe arise from independent experiments (fresh batch of portal protein and a unique solid-state nanopore).

In order to clarify this point we have altered the legend for figure 2e to read:
'(e) Conductance of solid-state nanopore vs conductance of hybrid SS-portal pores formed after electrokinetic 'corking' of the portal protein within the SS nanopore (n = 32 for CD/N hybrids and n = 15 for CGG hybrids).'

Regarding the number of events (ranging from 3,368 to 18,812 events) for each sensing experiment, please see the response to question 6 below.

6. Not clear what the number of events is for each experiment.

We thank the referee for this comment. We added in figure 4 legend the number of events for each biomolecule, which now reads:

'**Figure 4: Sensing different biopolymers using a hybrid nanopore.** Current vs time trace recorded through the hybrid pore at +60 mV in the presence of (a) 36.0 μ M insulin, (b) 7.7 μ M DNA hairpin, (c) 10.3 μ M TPX2 peptide and (d) 16.6 μ M ssDNA. The data in (a) were filtered at 10 kHz (grey) or 0.5 kHz (green). (e) Scatter plot of ΔI vs dwell time for the DNA hairpin (red, $n = 5883$ events), the peptide (purple, $n = 3368$ events) and the ssDNA (orange, $n = 18812$ events). Experiments were performed in 0.5 M NaCl, 20 mM Tris pH 7.5.'

7. Fig S7: some peaks appear biphasic- what is the cause?

As mentioned in the manuscript, we believe that the insulin is not translocated through the hybrid pore, because the dimensions of insulin do not allow its passage through the smaller

constriction of the portal protein. Therefore, insulin can interact unspecifically with the portal protein, and the biphasic events are likely transient explorations of an insulin molecule in the internal cavity of the pore, which has a 5 nm diameter that allows protein occupancy (see also Van Meervelt et al, JACS, 2017). Note that this figure has been renumbered to Figure S8 due to an additional figure inserted in response to a comment from reviewer 3, below.

8. According to the methods section, data was filtered at 10 khz; however, events often appear quicker than this which will result in artefacts in the dwell time and amplitude. This should be clarified or alternatively reanalysed.

We thank the referee for this comment. This study focused on the long-lived events in order to characterize transport rather than transient interaction. Similarly, we have focused our analysis on the longer events as they are more likely to represent translocations of the biomolecules (as shown for TPX2), rather than the fast events associated with transient binding without transport. The range of applied voltages (from 30 to 60mV) used in this study allows us to detect events from 276 μ s to the millisecond scale. As demonstrated before by Oukhaled et al, ACS Chem Biol, 2012, the amplitude of an event is merely affected by filtering at 10kHz. Therefore, in our case, filtering data at 10 kHz will not result in artefacts in the dwell time nor the amplitude.

Reviewer #3 (Remarks to the Author):

Recommendation: The supportive comments below suggest the reviewer recommends publication in Nature Comms with minor revisions.

Comments: The paper reports on the novel lipid-free hybrid nanopore-based on a natural DNA pore from a thermostable virus, electrokinetically inserted into a larger nanopore supported in a silicon nitride membrane. The authors present a carefully performed set of experiments, which are used to discuss the engineering and sensing properties of the hybrid nanopore. The information provided allows for replicating the experiments by other authors.

Unlike other hybrid pore system, authors claim to improve on the lifetime of the pores compared to the state of the art of hybrid nanopores (referring here to Hall et.all. 2010 Nature Nanotechnology) The robust geometry and improved noise performance of the hybrid system enabled sensing of several biomolecules TX2 peptide, ssDNA, harpin DNA and insulin.

I want to pinpoint a few remarks that are of importance to me and that the authors could help clarify.

1. The authors indicate at line 72 :“Our hybrid pores exhibit lifetimes of hours, and similar ion current noise values to a lipid bilayer supported portal protein nanopore³³ (Figure 2).” However, the trace is only 100 s long To support a lifetime of hours for hybrid pores it would be beneficial to prepare separate figure that shows a baseline of the same pore at the start of the experiment and at the end (several hours later).

We appreciate the suggestion, and as a result we have added an additional supplemental figure (Figure S4) to support this statement.

2. As discussed in the paper biological pores, due to their stable and reproducible geometry, have stable and reproducible current drops. Authors argue that their hybrid nanopore preserves this advantage of the biological pores. However, nanopores are corks to the ss-pores of different diameters (SI Figure 2 and 3). Diameters of ss-pores range from 5.9-5.4 nm, consequently as reported by the authors the level of leakage current, and noise were correlated with the size of the ss-pore.

Authors should comment more on how crucial is the size of ss-pore and suggest the strategy to compare the results from different hybrid pores. Ideally, one would like to take advantage and in the future perform high-throughput experiments. In the paper the authors should discuss how practical their approach would be in terms of high-throughput measurements.

We thank the referee for this comment. Indeed, the diameter of the solid-state nanopore is a crucial parameter. As mentioned in the manuscript “We find that stable insertion required a specific range of SS nanopore geometries to work well, namely, a diameter between 5.4 to 6 nm and a nominal membrane thickness of 30 nm”. We tested various solid-state nanopore diameters ranging from 5 to 8 nm. For diameters inferior to 5.4 nm, we never detected complete insertion of a portal protein, being too big to be corked. On the other end, for solid-state pores with a diameter superior to 7 nm, we often detected transient and unstable insertions, characterized by a high noise (see answer question 3 below).

We showed that the level of current blockade ΔI appears to be biopolymer dependent and is due to the translocation through the narrowest constriction of the hybrid pore. We also showed that they have a low level of constant peripheral leakage and that the reproducible signals obtained from biomolecules, as well as the steady baselines of the hybrid, allow current blockades as low as ~ 20 pA to be accurately measured. The current blockade ΔI could be used to compare different hybrid pores in high throughput experiments.

We agree with that geometrical definition afforded by the SS-nanopore platform is amenable to array based, or high-throughput measurements. As such we have altered the final sentence of the paper to read:

...‘produce robust and geometrically controlled devices *amenable to sophisticated high-throughput array based- or coupled multi-system- (eg. Optical, flow, or pressure)* applications in nanotechnology.’

3. SI. Figure 2 gives examples of portal pore CD/N insertions into the ss-nanopore for three different pore sizes, it is difficult to judge between the panels, but it looks like that the noise level is higher for the 5.5nm large ss-pore compared to panel c) ss-pore of 5.6 nm.

It is also unclear what is a difference between panels b and c) pore is 5.6 nm large and bias conditions are the same, but the noise level appears larger in b). Is this an identical pore, what changed during the experiment...

We thank the referee for this comment. Figure S2 represents 4 independent experiments of solid-state nanopores before and after insertion of a CD/N portal protein. Therefore, it means that for each experiment we used a different solid-state nanopore, unique by its geometry. Regarding the noise in panel b, we show that in some cases, portal proteins transiently insert into the solid-state nanopore (in this case for 20 seconds) and are not stable (high noise). This could be due to the geometry of the solid-state nanopore not allowing perfect corking of the portal, or simply incomplete corking permitting to the protein to jiggle inside the nanopore. On the other end, panel c represents complete and stable corking of the protein inside the solid-state nanopore characterized by a low noise after insertion. We believe that the characteristic noise after insertion is mainly due to the geometry of the pore which can be almost perfectly circular (low noise because the protein is well inserted and cannot fluctuate) or slightly oval (higher noise due to fluctuations of the protein inside the pore).

4. Due to the tremendous progress of ss-nanopore fabrication techniques based on dielectric breakdown and light assisted nanopore drilling, I was wondering if authors attempted the to cork CGG into ss-pores fabricated in that way?

How crucial is TEM since it appears as the preferred fabrication technique (ref. 58 and 59)?

For supporting information IVs and TEM images of the ss-pores would be beneficial as the small changes in the pore size have a significant impact on the hybrid performance. –judging from the SI Figure 2.

As recommended by the reviewer, we added in figure S3 two TEM images and their respective IVs. Indeed, insertion of the portal protein nanopore into SS nanopores fabricated using techniques other than TEM, such as laser-assisted dielectric breakdown, is the subject of current research.

REVIEWERS' COMMENTS:

Reviewer #3 (Remarks to the Author):

Authors addressed all my comments. The quality of the work is also generally very good and I recommend it for publication.